# Aeroelastic Optimization Design of the Global Stiffness for a Joined Wing Aircraft

**Xuyang Li [1], Zhiqiang Wan [1], Xiaozhe Wang [2],\* and Chao Yang [1]**

[1] School of Aeronautic Science and Engineering, Beihang University, Beijing 100191, China; lywwmm1998@sina.com (X.L.); wzq@buaa.edu.cn (Z.W.); yangchao@buaa.edu.cn (C.Y.)

[2] Institute of Unmanned System, Beihang University, Beijing 100191, China

\* Correspondence: wangxiaozhemvp@buaa.edu.cn

**Abstract:** Due to the complexity and particularity of the joined wing layout, traditional design methods for the global stiffness of a high-aspect wing are not applicable for a joined wing. Herein, a beam-frame model and a three-dimensional wing-box model are built to solve the global stiffness aeroelastic optimization design problem for a joined wing. The goal is to minimize the weight, and the constraints are the overall aeroelastic requirements. Based on a genetic algorithm, two methods for the beam-frame model and one method for the three-dimensional model are used for comparative analysis. The results show that the optimization method for a diagonal beam section and the optimization method for an exponential/linear combination function fit are adequate for optimizing and designating the joined wing global stiffness. The distributions obtained using the two methods have good consistency and are similar to the distribution of the three-dimensional model. The stiffness distribution data and the beam section parameters can be converted from each other, which is convenient for redesigning the structure parameters using the stiffness distribution data, and is valuable for engineering applications.

**Keywords:** joined wing; aeroelastic optimization; engineering beam theory; global stiffness design

## 1. Introduction

Aircraft design includes complex aeroelastic problems; therefore, it is necessary to use structural optimization techniques for a compromising primary design scheme, in order to solve issues related to the coupling between different disciplines, e.g., structural, aerodynamics, control, etc., and meeting aeroelastic performance requirements. A design under aeroelastic constraints is called an aeroelastic optimization. In the overall design of aeroelastic problems, the primary issue for determining the influence of structural deformation on the aerodynamic characteristics is designing a reasonable stiffness distribution for the wing structure [1], and this is also an essential basis for subsequent designs [2]. To obtain more accurate results that satisfy the aeroelastic performance requirements, it is necessary to design a wing stiffness distribution using design optimization technology, which can preferably guide the selection of the structural scheme and the arrangement of the macrostructure stiffness and mass distribution [3].

Joined-wing airplanes have been widely explored and studied in many different disciplines since Wolkovitch introduced the concept in 1976 [4]. A joined-wing airplane can be defined as an airplane that incorporates tandem wings arranged to form diamond shapes in both the plan and front views, and the fuselage can be seen as the connecting diagonal of the diamond frame. The advantages of a joined wing include light weight, high stiffness, low induced drag, good transonic area distribution, high-trimmed CLmax (maximum trimmed lift coefficient), reduced wetted area and parasite drag, direct lift control, side force control capability, excellent stability and controllability [5]. However, the connection between the front wing and the rear wing leads to dissimilarity in the structural and aerodynamic characteristics compared with a traditional layout, and the interconnected

wings form a complex overconstrained system, which increases the difficulty of analysis and increases the design space of different disciplines [6,7]. This leads to integrated design changes becoming the key problem, as the authors in [8] noted that basic aeroelastic investigations must be introduced early in the design process of a joined-wing aircraft, and a stiffness distribution design is also necessary.

Aeroelasticity research has been accompanied by the development of the joined wing. Robert A. Canfield and his team performed a series of studies on the aeroelasticity of a joined wing during the first decade of the 21st century. Different structural models of joined wings were studied, including high-fidelity finite elements method (FEM)-based weight models [8], embedded antenna models [9,10], nonlinear structure models [11,12], sensor craft models [13,14], scale models [15–18], a beam model [19], and a wind tunnel model [20]. Nearly half of the research dealt with linear structural integrated design and optimization, whereas the other half dealt with the nonlinearity of the joined-wing models. Demasi et al. also performed great research on the aeroelasticity of joined wings. Their research focused mainly on the nonlinear problems of joined wings. The evaluation of aeroelastic characteristics of the joined wing was undertaken in the early design stages in [21]; both reduced-order models [22,23] and full-order models [24] were used to analyze the nonlinearity of joined wings; and post-critical phenomena were researched in [25,26]. However, the aeroelastic optimization of the models has hardly been taken into account. Preliminary analyses on joined wings showed that aerodynamic loads at the tip of the wing were sensitive to the modeling with reference to efforts [27,28]; moreover, aeroelastic analyses showed large differences in the predicted flutter speeds. Thus, it was speculated that by attributing the forces in the tip regions largely to the bending moment, the overall aeroelastic response was significantly affected by the redistribution. Flutter involving vehicle motion can be an active constraint [29].

As mentioned above, the aeroelastic analysis of a joined wing is necessary, and as stated earlier, optimization in the preliminary stage of aircraft design is also indispensable; therefore, the aeroelastic optimization of a joined wing is important. A large number of aeroelastic optimizations of joined wings have been explored in precursory studies. The rib thicknesses of a scaled joined wing was studied as a variables, and the scaled natural frequencies and scaled flutter speed of the full-scale vehicle were reproduced after optimization in [14]. An equivalent unsymmetrical beam section was used for structural optimization and was proven to adequately approximate the stiffness and deflection behavior of a real wing in [30]. Aerostructural optimization was carried out in [31,32]. A panel method and an equivalent beam finite-element model were used for aerostructural optimization with the aerodynamic design variables as the parameters of the geometric configuration, and the thicknesses of the spar wall were added as structural design variables. The application of bacterial foraging optimization to a joined wing was shown in [33] and a hybrid variant was applied to match the aeroelastic responses of a wind tunnel model with a full-scale aircraft.

Compared with traditional airplanes, structural optimization and weight estimations for the preliminary design of a joined-wing aircraft lack mature methods. In engineering, traditional wing-box design procedures and empirical formulas for traditional wing stiffness distributions have been frequently used for the preliminary design of a joined wing. However, this approach has three problems. Firstly, as it is a complex overconstrained system, similar external geometric parameters in the layout lead to completely different thicknesses of the internal structures and thus weights. Secondly, to minimize the weight by fixing the topology of the wing box, the material needs to be redistributed differently from what is usually seen in common layouts. As a result, the secondary bending moment tilts the neutral axis, and thus material is more efficiently utilized when distributed far apart to the two opposite corners. Thirdly, because of the interaction of the front wing and the rear wing, the distribution of the bending moment of the joined wing is different from traditional wings; thus, the distribution of the stiffness is different. For these reasons,

optimization approaches in regard to stiffness based on traditional wings are not applicable to joined wings.

In the conceptual and preliminary design stages, there are many uncertain parameters. Stiffness design requires a large amount of relevant data, such as airfoil, load, structural layout, etc., and this information is often obtained through parallel or subsequent design processes. This lack of information will lead to a decrease in the accuracy of the stiffness design, which will require repeated modifications of the structural stiffness distribution in the subsequent design, resulting in a waste of manpower and time. Furthermore, parametric analysis of the joined wing is even more difficult due to its complex layout compared with the traditional airplane. This leads to the low design efficiency of the detailed finite-element models and nonlinear structure models. Therefore, it is necessary to introduce a simple stiffness model that can describe joined-wing structural characteristics directly and efficiently, so as to provide guidance for the subsequent structural scheme screening, macro-structural layout, and mass distribution.

It can be seen from the above that there are few direct studies on the stiffness distribution of joined-wing structures; however, it is necessary to conduct an in-depth study in the preliminary design stage, while considering the aeroelastic problem. In order to solve this problem, this paper introduces three different aeroelastic optimization methods in the preliminary stage to design the stiffness of the joined wing.

## 2. Materials and Methods

### 2.1. Beam Cross-Section Simplification

As shown in Figure 1, the out-of-plane components of a joined wing aircraft tend to bend the wings about a tilted bending axis. To resist this, the wing's structural material must form a deep spar about this axis. This implies that the material must be concentrated near the upper leading edge and the lower trailing edge. As shown in Figure 2, to simulate the real material distribution of the beam section of a joined wing, three section shapes are simplified to facilitate calculation without losing rationality. Through a series of theoretical derivations, the best scheme for a joined wing is selected from the three section shapes. As shown in Figure 2a, section parameters $X_1$–$X_4$ are used for the design parameters in each section of the beams. The parameter $X_1$ represents the main diagonal flange size; $X_2$ represents the secondary diagonal flange size; $X_3$ represents the web thickness; $X_4$ represents the skin thickness; and $X_5$ and $X_6$ represent the width and height of the wing box, respectively, and they are fixed in each beam section. The symbol $\alpha$ represents the tilted angle, as shown in Figures 1 and 2.

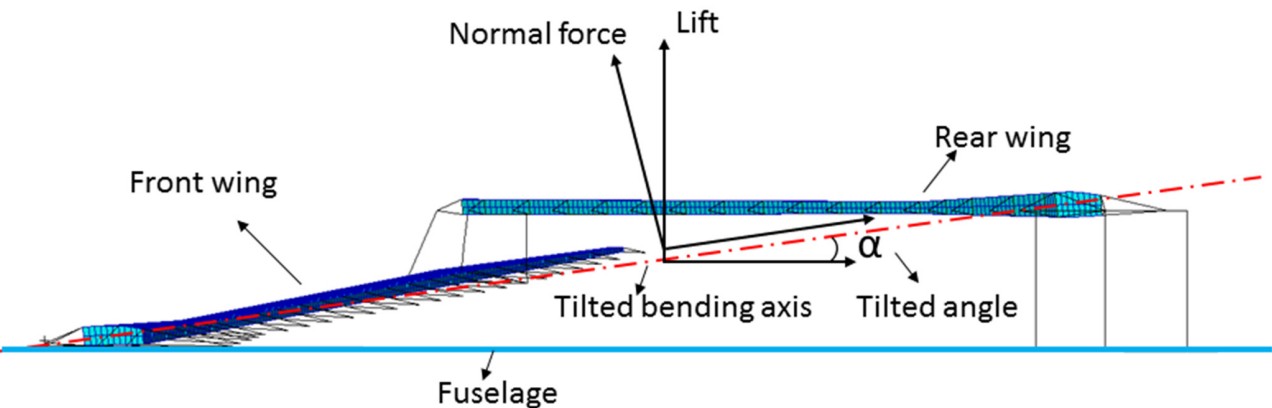

**Figure 1.** The structural layout, forces, and tilted bending axis of a three-dimensional joined-wing model.

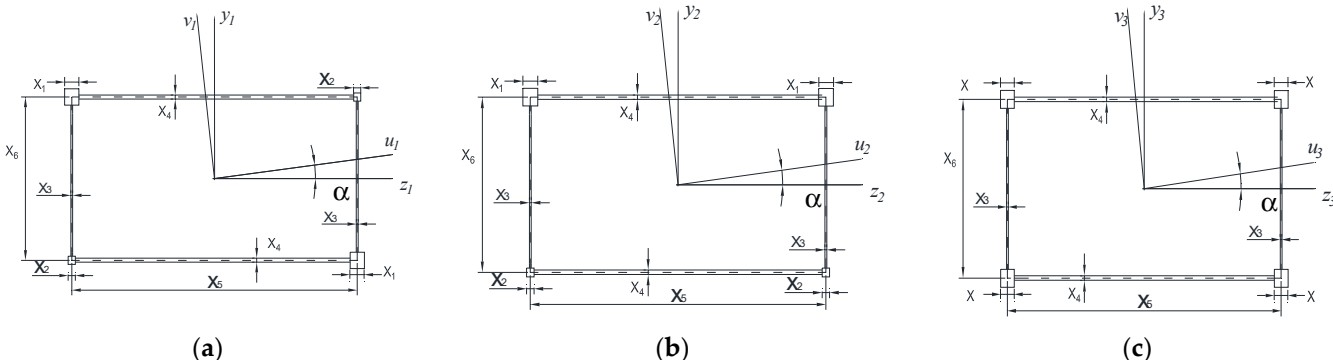

**Figure 2.** Simplified beam sections with a tilted bending axis. (**a**) Centrosymmetry beam section; (**b**) axial symmetry beam section; (**c**) same diagonal beam section.

As shown in Figure 2a, the equations used to evaluate the principal inertia moments $I_{z1}$ and $I_{y1}$, the torsional stiffness coefficient $I_{t1}$, and the inertial product $I_{yz1}$ of the cross section are as follows:

$$I_{z1} = \frac{1}{2}A_t X_6^2 + \frac{X_3(-\overline{X}^3 - 2X_6^3 + 3X_6\overline{X}^2)}{6} \tag{1}$$

$$I_{y1} = \frac{1}{2}A_t X_5^2 + \frac{X_4(-\overline{X}^3 - 2X_5^3 + 3X_5\overline{X}^2)}{6} \tag{2}$$

$$I_{t1} = \frac{4\Omega^2}{\oint \frac{ds}{\delta}} \approx \frac{2X_3 X_4 (X_5 X_6)^2}{X_4 X_6 + X_3 X_5} \tag{3}$$

$$I_{yz1} = \frac{X_5 X_6}{2}(A(X_2) - A(X_1)) \tag{4}$$

where:

$$A(X_1) = X_1^2, \quad A(X_2) = X_2^2, \quad A(X_3) = X_3 L_w, \quad A(X_4) = X_4 L_s$$
$$A_t = \sum_{i=1}^{4} A(X_i), \quad L_s = X_5 - \overline{X}, \quad L_w = X_6 - \overline{X}, \quad \overline{X} = \frac{X_1 + X_2}{2} \tag{5}$$

In general, the flange size is larger than the size of the web thickness and skin thickness, and as mentioned earlier, to simulate the material distribution, the main diagonal flange size is larger than the secondary diagonal flange size shown in Figure 2a, so the relations of the cross section parameters are as in Equation (6):

$$\max(X_3, X_4) < X_2 < X_1 \tag{6}$$

The section area can be calculated as:

$$S = 2A_t \tag{7}$$

According to the rotation axis formula of the moment of inertia in Equation (8), the moment of inertia of a beam section relative to the bending axis $u_1$ in Figure 2a can be derived as Equation (9):

$$I_u = \frac{I_z + I_y}{2} + \frac{I_z - I_y}{2}\cos 2\alpha - I_{yz}\sin 2\alpha \tag{8}$$

$$I_{u1} = \left(1 - \alpha^2\right)I_{z1} + \alpha^2 I_{y1} - 2\alpha I_{yz1} \tag{9}$$

For the case of Figure 2b:

$$I_{z2} = I_{z1} \quad I_{y2} = I_{y1} \quad I_{yz2} = 0 \tag{10}$$

then

$$I_{u2} - I_{u1} = 2\alpha I_{yz1} < 0 \tag{11}$$

For the case of Figure 2c, the four flange areas are equal. Defining the flanges $X$ (where $X = X_1 = X_2$) as variables and keeping parameters $X_3$–$X_6$ and $A_t$ unchanged, according to Equations (3)–(5):

$$I_{z3} = \frac{1}{2} A_t X_6^2 + \frac{X_3(-X^3 - 2X_6^3 + 3X_6 X^2)}{6} \tag{12}$$

$$I_{y3} = \frac{1}{2} A_t X_5^2 + \frac{X_4(-X^3 - 2X_5^3 + 3X_5 X^2)}{6} \tag{13}$$

$$I_{yz3} = 0 \tag{14}$$

$$I_{u3} = \left(1 - \alpha^2\right) I_{z3} + \alpha^2 I_{y3} - 2\alpha I_{yz3} \tag{15}$$

By a series of derivations, as shown in Appendix A Equations (A15)–(A19), the result is:

$$\Delta I_u = I_{u1} - I_{u3} = X_6(\alpha X_5(X_1^2 - X_2^2)) + X_3(\overline{X}^2 - X^2) \tag{16}$$

By a series of derivations, as shown in Appendix A Equations (A20)–(A25), when the Equation (17) condition is met, the result $\Delta I_u > 0$ holds.

$$N > \frac{(1 - \frac{k_2^2}{4})(k+1)}{(k-1)k_2^2 \alpha} \tag{17}$$

where:

$$N = \frac{X5}{X3}, \ k = \frac{X1}{X2} > 1, k_2 = \frac{X3+X4}{2X} \tag{18}$$

That is, if the above formula is satisfied, the moment of inertia relative to the bending axis of the case in Figure 2a is greater than that in Figure 2c when the section areas are the same. For thin-walled wing boxes, the above formula is generally satisfied, which means that when the areas of the sections are the same, i.e., when the weights of the beams are the same, the section in Figure 2a has a larger resistance to the tilted bending axis. In other words, a joined wing using the simplified beam section in Figure 2a requires less weight when the stiffness is the same; therefore, the section layout in Figure 2a is the best scheme in theory, so it is used for optimization.

## 2.2. Variable Model Division

Three methods are applied to solve the problem of the aeroelastic optimization of a joined wing, including two methods for a beam-frame model and one for a three-dimensional model, which are used for comparative analysis. The division of the variables of the models are described in detail below.

As shown in Figure 3, the joined wing is divided into three partitions, i.e., the front wing, the outer wing, and the rear wing. The main beam of each partition is then subdivided into several sections. More specifically, as shown in Figure 4 in different colors, there are three design sections between the fuselage and the kink point both in the front wing and the rear wing and six design sections between the kink point and the joint. The outer wing is divided into four design sections. For each section of the beam of the beam-frame model, two methods are used to address its stiffness, namely, method A and method B (The two methods will be discussed in detail in Section 2.3).

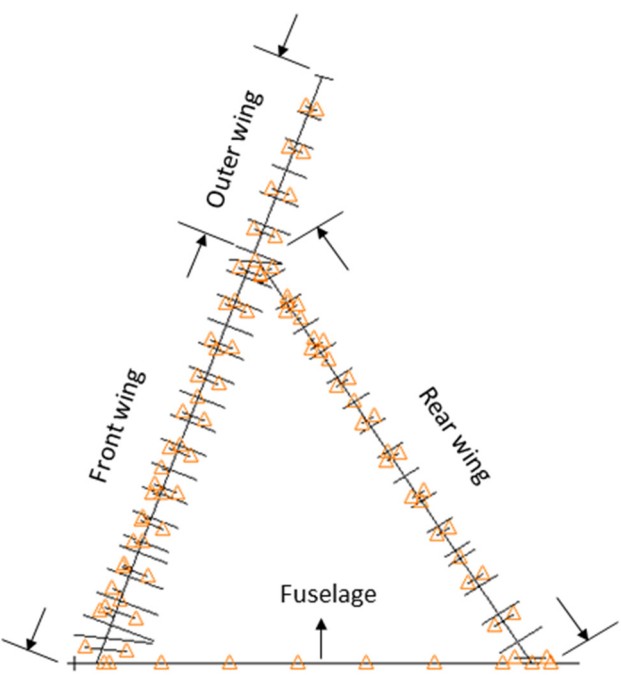

**Figure 3.** Partitions of a joined wing beam model of 3 partitions, i.e., the front wing, the outer wing, and the rear wing.

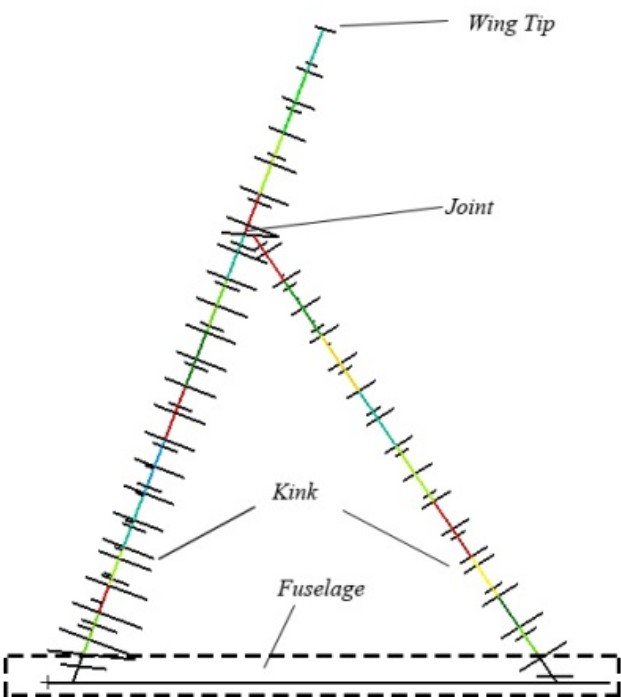

**Figure 4.** Sketch of the beam-frame wing of different design sections.

Method C is a traditional method to solve aeroelastic optimization problems with detailed scales of structure, and it is applied to the three-dimensional model, as shown in Figure 5. The wing box is divided into several sections in the same way as the beam-frame model, as shown in Figure 6. The difference is that the design variables of this method are the real structural sizes of the wing box, e.g., the thicknesses of skins, webs, and flanges.

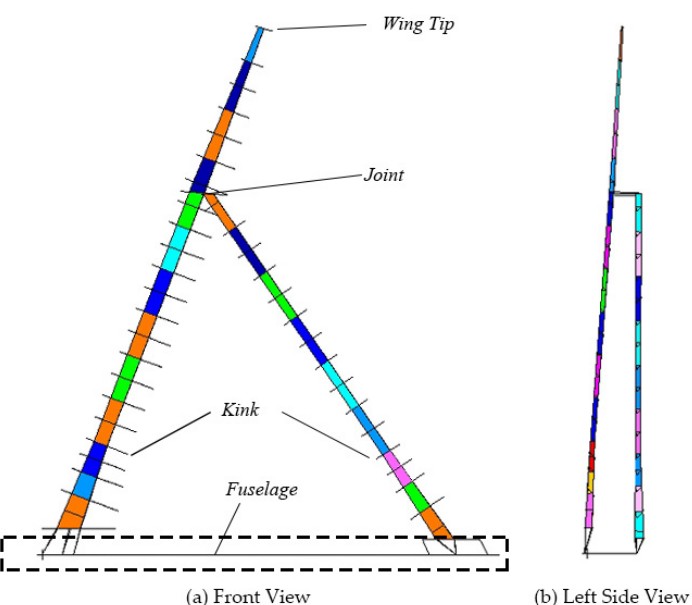

**Figure 5.** Sketch of the three-dimensional wing of different design sections.

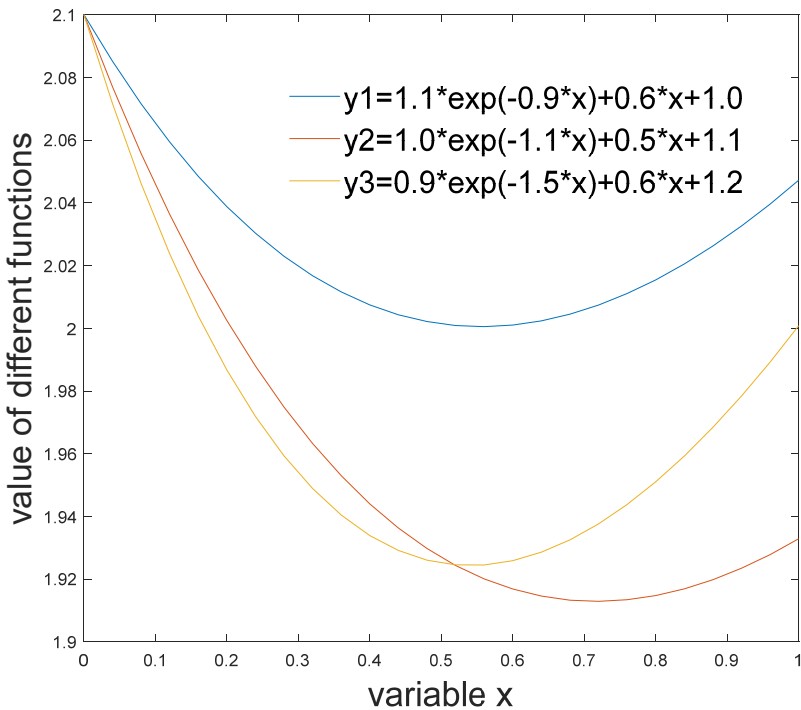

**Figure 6.** Sketch of exponential-linear mixed functions with different parameters. (The parameters of these functions are given arbitrarily to show the influence of different parameter changes on the trend of the functions.).

### 2.3. Aeroelastic Optimization of the Global Stiffness

An aeroelastic optimization study is concerned with a standard optimization problem, which minimizes an objective function subject to constraints that limit the values of the design variables in $n_d$ dimensional space. The problem can be written as [34]:

$$g_j(v) \leq 0 \, (j = 1, 2, \cdots, n_c) \tag{19}$$

$$(v_i)_{\text{lower}} \leq v_i \leq (v_i)_{\text{upper}} \, ( \, i = 1, 2, \cdots, n_d) \tag{20}$$

where $g_j(v)$ is a correlation function of the constraint index; $v$ is a vector set of the design variables; $v_i$ is a single design variable; $(v_i)_{\text{lower}}$ is the lower bound of the design variables; $(v_i)_{\text{upper}}$ is the upper bound of the design variables; $n_c$ is the number of constraints; and $n_d$ is the number of design variables. In aeroelastic design problems, the objective function $F(v)$ is generally mass, that is, to find the set of design variables that meet the conditions $v_i$ to minimize the total mass of the structure. The constraint conditions are generally static aeroelastic and dynamic aeroelastic constraint indexes, such as flutter speed, aileron efficiency, structural deformation and stress, which are constrained by Equation (19). The upper and lower limits of each design variable are constrained by Equation (20).

An optimization algorithm based on the basic genetic algorithm is introduced for aeroelastic optimization problems [2].

### 2.3.1. Method A

As shown in Figure 2a, section parameters $X_1$–$X_4$ are used for the design parameters in each section of the beams. Let the main diagonal flange size be $X_1$, the secondary diagonal flange size be $X_2$ ($X_1 > X_2$), the web thickness be $X_3$, and the skin thickness be $X_4$. The width and height of the wing box are $X_5$ and $X_6$, respectively. The section parameters $X_3$ and $X_4$ of each section of the beam and the overall ratios $K_1$ and $K_2$ are the design variables as follows:

$$\begin{aligned} K_1 &= X_1 / X_2 \\ K_2 &= 2X_2 / (X_3 + X_4) \end{aligned} \tag{21}$$

There are 40 total design variables. Section parameters $X_5$ and $X_6$ are invariant in each section but vary with the spanwise direction of the beams. Generally, the overall ratios are larger than 1. In particular, when we make the design variable $K_1$ equal to 1, the situation then changes into a beam section, as shown in Figure 2c.

### 2.3.2. Method B

Referring to the stiffness distribution law of the traditional high-aspect-ratio wing, the stiffness distributions along the spanwise direction of the front wing, outer wing, and rear wing are optimized according to an exponential function as follows [2]:

$$GJ(y), EI(y) = ae^{by} + c \tag{22}$$

A set of three design variables (i.e., $a$, $b$, and $c$) can be defined for each kind of stiffness. Thirty-six design variables are required because the vertical bending, horizontal bending, torsional stiffness, and stiffness caused by section asymmetry of each partition need to be designed.

Due to the difference between a joined-wing shape and a traditional high-aspect-ratio wing, a traditional exponential stiffness distribution is not appropriate for a joined wing. As noted in previous papers, the moment of a joined wing along the spanwise direction decreases first and then increases in the middle part. To have a lighter weight, the distribution should have the same trend. Therefore, a new function needs to be found to simulate the distribution of a joined wing. As shown in Figure 6, an exponential/linear mixed function can change with its parameters and simulate the trend of the stiffness of a joined wing well, so it is used to modify the distribution as follows:

$$GJ(y), EI(y) = Ae^{By} + C + Dy \tag{23}$$

A set of four design variables (i.e., $A$, $B$, $C$, and $D$) can be defined for each kind of stiffness, similarly to method B. Equation (23) is used to address the optimization problem because including a linear term $Dy$ can simulate the transition of the stiffness curve of a joined wing well. Forty design variables are required because the vertical bending, horizontal bending, torsional stiffness, and stiffness caused by the section asymmetry of each partition need to be designed.

### 2.3.3. Method C

A three-dimensional model, as shown in Figure 7, is employed. The model is divided into the same three parts as the previous beam-frame model, i.e., the front wing, outer wing, and rear wing. The thickness values of the flanges, webs, and skins in each partition are used individually for the aeroelastic optimization using a genetic algorithm. The details of this method are presented in [2].

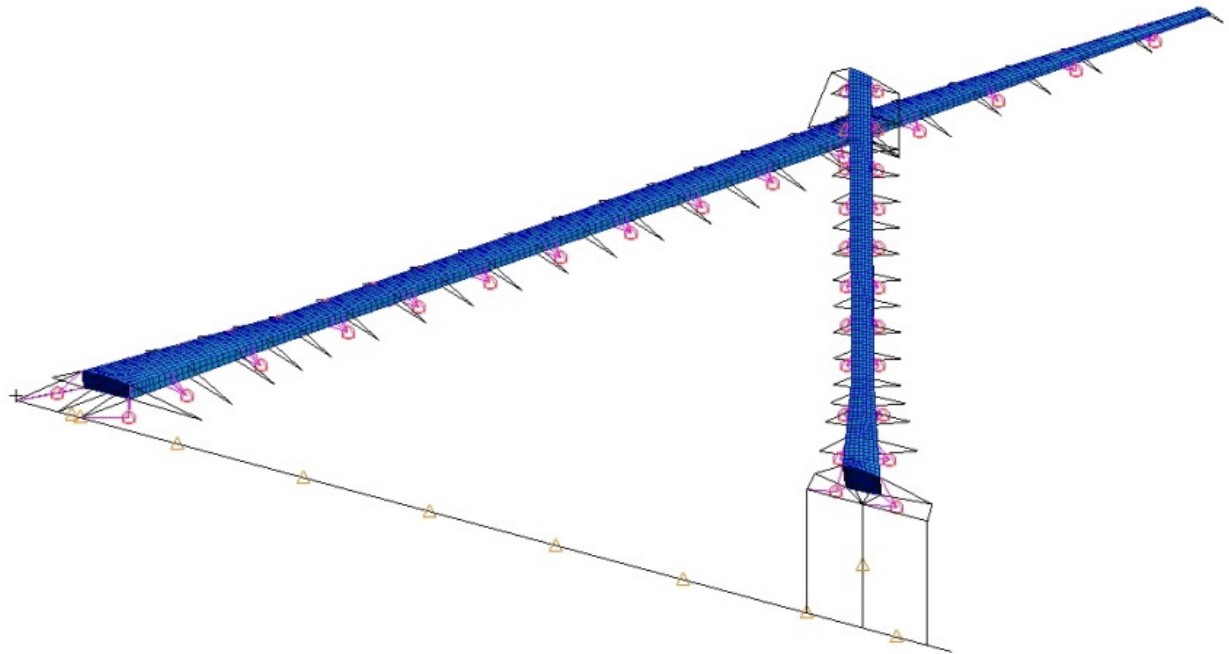

**Figure 7.** Three-dimensional wing-box model of a joined wing.

### 2.4. Aeroelastic Responses

The methods that deal with the aeroelastic responses involved in this paper are described as follows.

#### 2.4.1. Equilibrium Equation for Static Aeroelastic Response Analysis

In general, the static aeroelastic equation is established under the displacement vector $f - set$, expressed as [35]:

$$[K_{ff} - \bar{q}Q_{ff}]u_f + M_{ff}\ddot{u}_f = \bar{q}Q_{fx}u_x + P_f \tag{24}$$

where $K_{ff}$ represents the structural stiffness matrix; $\bar{q}$ indicates the dynamic pressure; $u_f$ represents the displacement vector; $M_{ff}$ represents the structural mass matrix; $M_{ff}\ddot{u}_f$ represents the inertia force caused by the rigid body motions; $P_f$ represents the external load vector; $\bar{q}Q_{ff}u_f$ represents the aerodynamic increment caused by structural elastic deformation; and $\bar{q}Q_{fx}u_x$ represents the aerodynamic force caused by rudder deflection and rigid body motion of the aircraft. Superscript ".." is the quadratic derivative of time.

#### 2.4.2. Flutter Function

There are generally three common flutter analysis methods, the $V - g$ method and $p - k$ method and the matrix eigenanalysis method [36], which are all computationally

efficient and suitable for flutter analysis in optimization problems. The $p - k$ method is used in this paper. The equation is established under the modal set $h$ [37]:

$$\left[\left(\frac{V}{b}\right)^2 p^2 M_{hh} + \frac{V}{b} p B_{hh} + K_{hh} - \frac{1}{2}\rho V^2 \left(Q_{hh}^R + \frac{p}{k}Q_{hh}^I\right)\right]u_h = 0 \tag{25}$$

where $V$ is the incoming flow velocity; $b$ is the reference half chord length; $p$ is the eigenvalue; $k$ is the reduced frequency; $M_{hh}$ is the modal mass matrix; $B_{hh}$ is the damping matrix; $K_{hh}$ is the modal stiffness matrix; $Q_{hh}$ is the aerodynamic force matrix; and the superscripts $R$ and $I$ indicate the real and imaginary parts, respectively.

## 3. Model Description

### 3.1. Beam-Frame Structure Model

A half-model of a high-aspect-ratio joined wing is used as the research object, and detailed design parameters of the wing are given in Table 1. The structural model of the joined wing in the preliminary design stage is shown in Figure 3. It is composed of 270 nodes and 367 elements, including 93 mass point elements and 274 rod elements, which is established using Patran. The wing stiffness is simulated by an elastic axis from the wing root to the wing tip, and the mass characteristics are simulated by the lumped mass. An equivalent material was used by referring to the average values for aluminum alloys for the main beams of the wings. Its density $\rho_{alu}$ = 2700 kg/m$^3$, Young's modulus $E$ = 72 Gpa, and Poisson's ratio $\mu$ = 0.33. All structures except the beams, e.g., the fuselage, ribs, and the fin and joint plate, are rigid.

**Table 1.** Aeroelastic constraint conditions.

| Constraints | $d_{t,z}$ | $d_{t,x}$ | $d_{j,z}$ | $d_{j,x}$ | $\phi_t$ | $\phi_j$ | $V_f$ |
|---|---|---|---|---|---|---|---|
| Upper limit | 7.5% × $lt$ | 1.5% × $lt$ | 7.5% × $lj$ | 1.5% × $lj$ | 2° | 2° | - |
| Lower limit | - | - | - | - | −2° | −2° | 90 (m/s) |

where $d_{t,z}$ is the vertical deformation of the wing tip, $d_{t,x}$ the horizontal deformation of the wing tip, $d_{j,z}$ is the vertical deformation of the joint, $d_{j,x}$ is the horizontal deformation of the joint, $\phi_t$ is the twist angle of the wing tip, $\phi_j$ is the twist angle of the joint, $V_f$ is the constraint flutter speed, $l_t$ is the half wing span, and $l_j$ is the length between the wing root and the joint.

### 3.2. Three-Dimensional Structure Model

The wing is divided into panels, spars, ribs, and beams based on structural arrangement and dimension data. The three-dimensional structure model is shown in Figure 7. In accordance with the load-bearing characteristics of the various components, the entire wing is simulated by rod-shell elements. The upper/lower panels, wing ribs, and webs of the front/rear beams are simulated by shell elements. The spars and flanges of the front/rear beams are simulated by rod elements. The body, joint, and vertical tail are simulated by rigid beams. Although a large number of elements means that the results are more accurate, a moderate number of elements is more suitable for parameterization and optimization in the conceptual and preliminary stages, considering time-consumption. The entire wing model is composed of 8247 finite elements, which is sufficiently accurate and efficient in the current design stage.

### 3.3. Aerodynamic Model

The geometric shape parameters are shown in Figure 8, and the values of the parameters are shown in Table 2. A flat aerodynamic mesh considering curvature and the double-lattice method was applied to the aerodynamic model, as shown in Figure 9. The total area of the half-model is 72.76 m$^2$, among which the front wing is 46.51 m$^2$, whereas the rear wing is 26.25 m$^2$.

**Table 2.** Values of the shape parameters of the joined-wing layout.

| Shape Parameter | Value (m) | Shape Parameter | Value (m) | Shape Parameter | Value (°) |
|---|---|---|---|---|---|
| $l$ | 22.86 | $b_r$ | 2.54 | $\chi_f$ | 22.0 |
| $l_k$ | 4.63 | $b_{rk}$ | 1.49 | $\chi_r$ | 35.0 |
| $l_j$ | 15.67 | $b_{rj}$ | 1.49 | $\varphi_f$ | 4.0 |
| $b_f$ | 3.51 | $b_t$ | 0.75 | $\varphi_r$ | 0.0 |
| $b_{fk}$ | 2.11 | $h$ | 1.19 | | |
| $b_{fj}$ | 2.11 | | | | |

$l$ is the half span of the joined wing; $l_k$ is the length between the wing root and the kink; $l_j$ is the length between the wing root and the joint; $b_f, b_{fk}, b_{fj}$ are the chord lengths of the wing root, kink, and joint of the front wing, respectively; $b_r, b_{rk}, b_{rj}$ are the chord lengths of the wing root, kink, and joint of the rear wing, respectively; $b_t$ is the chord length of the wing tip; $h$ is the height of the joint plate; $\chi_f$ is the leading back swept of the front wing and outer wing; $\chi_r$ is the edging forward swept of the rear wing; $\varphi_f$ is the dihedral of the front wing and outer wing; $\varphi_r$ is the dihedral of the rear wing.

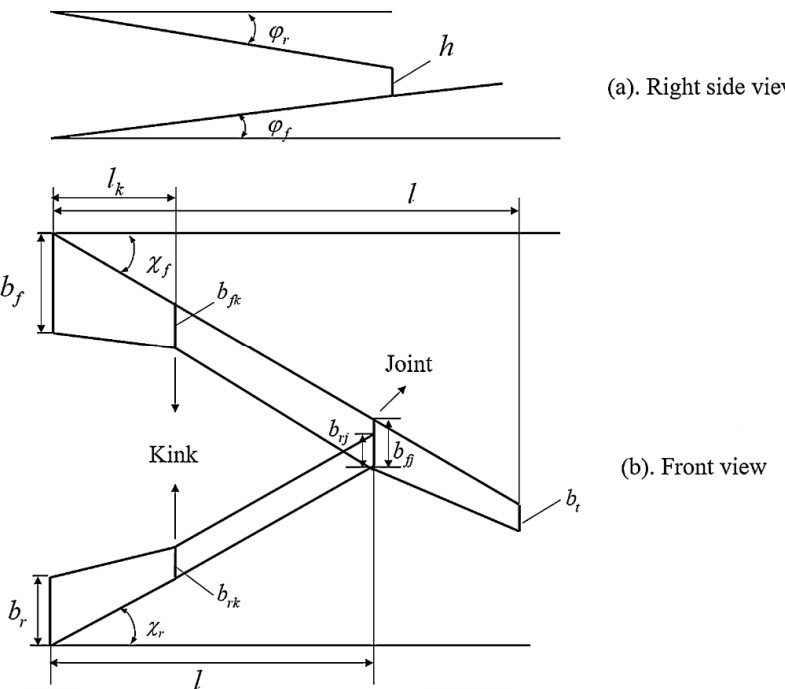

(a). Right side view

(b). Front view

**Figure 8.** Shape parameters of the joined-wing layout.

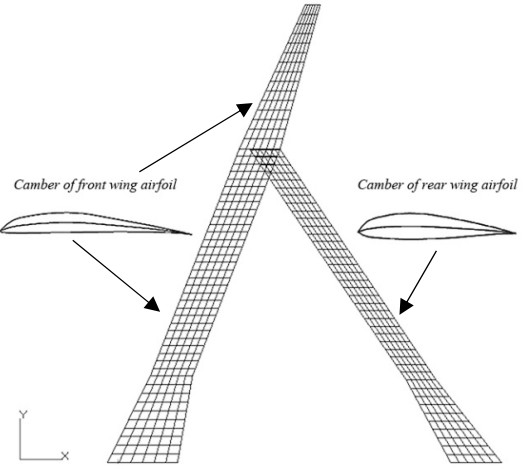

**Figure 9.** The double-lattice flat aerodynamic model considering curvature. The left camber is used for the front wing and the outer wing; the right camber is used for the rear wing.

## 4. Results and Discussion

### 4.1. Aeroelastic Responses of the Optimized Models

The constraint conditions of all the methods are the same, including the static aeroelastic responses and flutter constraint. The specific constraint values are shown in Table 1.

The displacement and torsion angle of the wingtip and the joint are the most concerning static aeroelastic responses, and a comparison of those values is given in Table 3. The flight condition is defined at $Ma$ = 0.2, a 1.5 g overload and 25,000 m cruise altitude, where the acoustic velocity $V_a$ = 298.4 m/s, the air density $\rho_{air}$ = 0.04 kg/m$^3$. The cruise speed $V_{cr}$ = 59.7 m/s, and the dynamic pressure $q$ = 71.28 Pa. The aeroelastic deformation of the beam-frame model optimized using method A is shown in Figure 10. The deformation of the other beam-frame model optimized using method B is similar to what is shown in Figure 10, so it is not included to avoid repetition. The aeroelastic deformation of the 3-D model optimized using method C is shown in Figure 11. Figure 12 depicts one of the $V - g$ and $V - f$ diagrams obtained based on the $p - k$ method. The other flutter result diagrams are not included to avoid repetition. All the flutter speeds of the optimized models are larger than the constraint speed. The flutter constraint $V_f$ is larger than 90.0 m/s, which seems very low; however, as the designed cruise speed $V_{cr}$ is only 59.7 m/s, and $V_f$ is larger than 1.5 times $V_{cr}$, the constraint speed is reasonable.

**Table 3.** Aeroelastic responses of the optimized models.

| Responses | Method A | Method B | Method C | Original |
|:---:|:---:|:---:|:---:|:---:|
| $d_{t,z}/l_t$ | 7.49% | 7.50% | 7.50% | 13.2% |
| $d_{t,x}/l_t$ | 1.30% | 1.15% | 0.30% | 2.23% |
| $d_{j,z}/l_j$ | 5.64% | 5.63% | 6.48% | 10.1% |
| $d_{j,x}/l_j$ | 1.09% | 1.01% | 0.41% | 2.05% |
| $\phi_t/°$ | 0.99 | 0.50 | 0.08 | 1.74 |
| $\phi_j/°$ | −0.05 | −0.26 | −0.64 | −1.43 |
| $V_F$ | >90 (m/s) | >90 (m/s) | >90 (m/s) | >90 (m/s) |

"Method A", "Method B", and "Method C" represent the aeroelastic responses of the models optimized using different optimization methods, respectively; "Original" represents the aeroelastic responses of the original model; $V_F$ is the actual flutter speed.

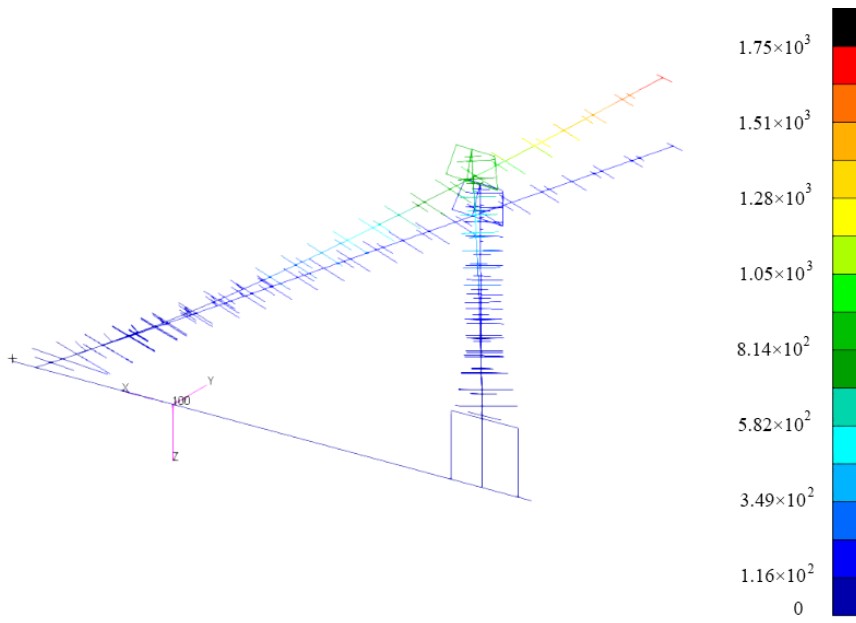

**Figure 10.** Aeroelastic deformation nephogram of the joined wing optimized using method A under a 1.5 g overload.

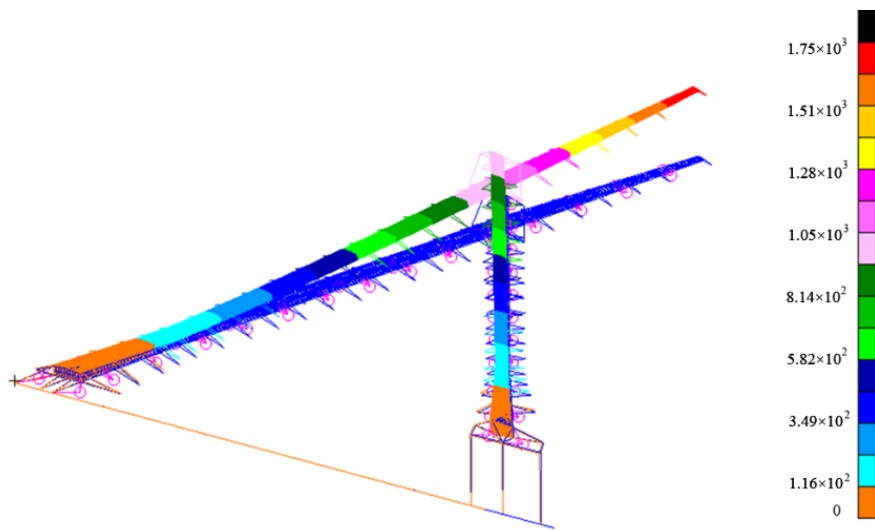

**Figure 11.** Aeroelastic deformation nephogram of the joined wing optimized using Method C under a 1.5 g overload.

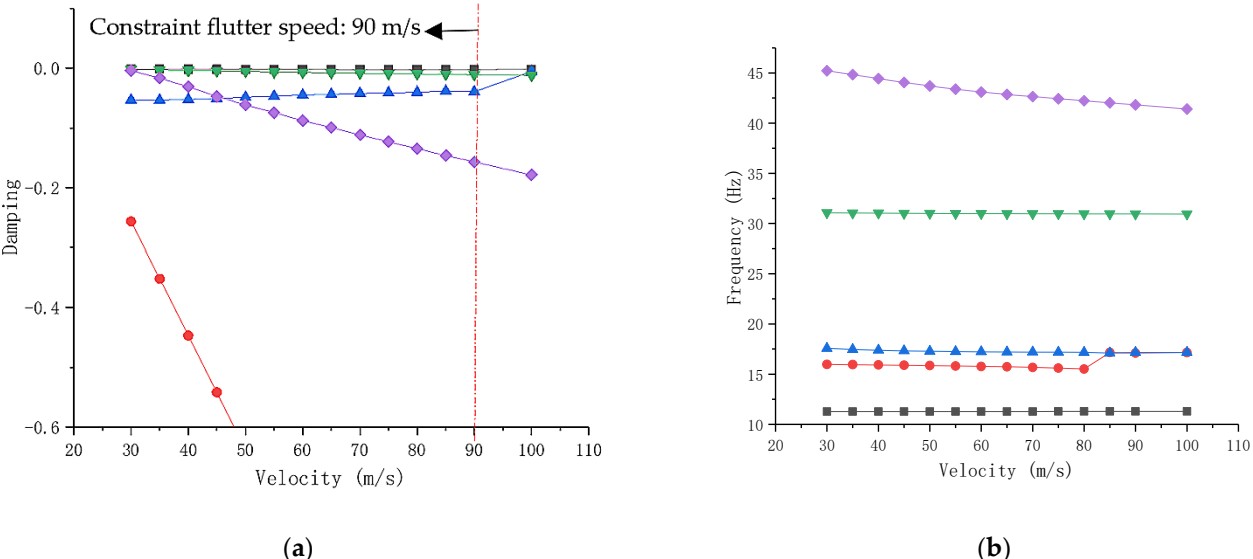

(**a**)　　　　　　　　　　　　　　　　　　　　　　　(**b**)

**Figure 12.** Flutter results of the optimized model at a constraint flutter speed of 90 m/s. (**a**) $V-g$ diagram (velocity-damping diagram); (**b**) $V-f$ diagram (velocity-frequency diagram).

The specific data of the aeroelastic constraints are listed in Table 3. From the table, we can see that compared with the original model, the constraints are satisfied well after optimization by all the methods, with only the vertical deformation of the wingtip reaching the constraint boundary value, and the other constraints having significant margins, so the optimization result is reasonable. It should be noted that the horizontal deformation of the wingtip is not negligible.

### 4.2. Results of the Optimized Parameters

The width and height of the beam sections, i.e., $X_5$ and $X_6$ respectively, are fixed and presented in Figure 13. The span station is nondimensional. The front wing and the outer wing are continuous in the span direction and the span station is from the wing root to the wing tip, whereas the span station of the rear wing is from the wing root to the joint. The descriptions of the nondimensional span station in later figures are the same.

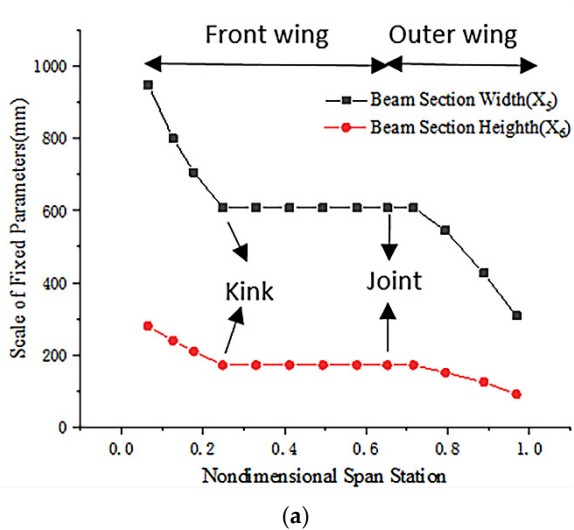
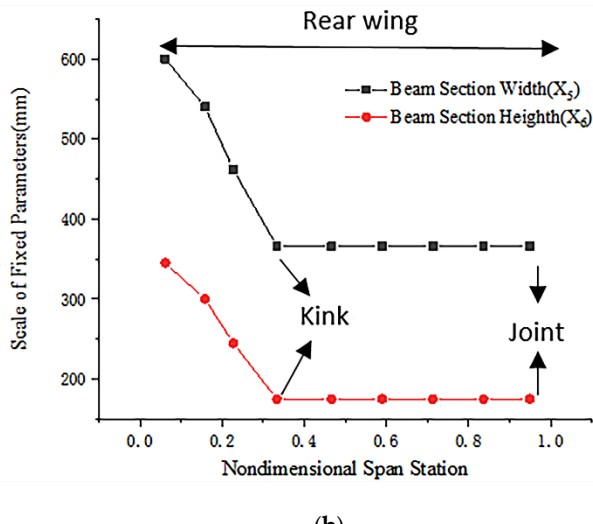

**Figure 13.** Scale of the fixed parameters along the span direction. (**a**) Scale of the fixed parameters along the span direction of the front wing and the outer wing; (**b**) scale of the fixed parameters along the span direction of the rear wing.

According to Equations (A3)–(A7), the parameters of the beam sections, i.e., $X_1$–$X_6$, are positively correlated with the torsional stiffness coefficient $I_{t1}$ and inertias $I_{z1}$ and $I_{y1}$. Because the scales of the fixed parameters, i.e., $X_5$ and $X_6$, are far greater than the design parameters, i.e., $X_1$–$X_4$, the inertias are primarily determined by the fixed parameters, so when the fixed parameters change, the inertias have the same trend; thus, we mainly consider the middle parts between the kink and the joint point where the fixed parameters are invariant with the span direction.

All the optimizations were completed in 20 generations, and there were 600 individuals in each generation. In theory, as mentioned in Section 2.1, the diagonal beam section is the best-simplified beam section for the stiffness optimization of a joined wing. As shown in Figure 2a, the parameters of each beam section are designed and optimized using a genetic algorithm in method A. The beam section parameters are solved in a reverse fashion using Equations (1)–(9) in method B. On one side, the areas of the beam sections are calculated, then the volume and mass of the beam are considered the objective, and on the other side, those beam section parameters can be used as a reference for a real scale of the beam sections, which is convenient for redesign work in the primary stage of joined wing design. The parameters of method C are the scales of the skins, flanges, and webs, which are in accordance with method A.

Figures 14 and 15 show the beam section parameters of the front wing and the rear wing, respectively. Based on these results, we can see that all four parameters of the three methods have the same tendency, i.e., in the inner part before the kink of both the front wing and rear wing, these parameters gradually increase with the span direction; then, in the middle part between the kink and the joint point of both the front wing and rear wing, these parameters first decrease and then increase up to the joint point; finally, in the outer part, they gradually decrease with the span direction.

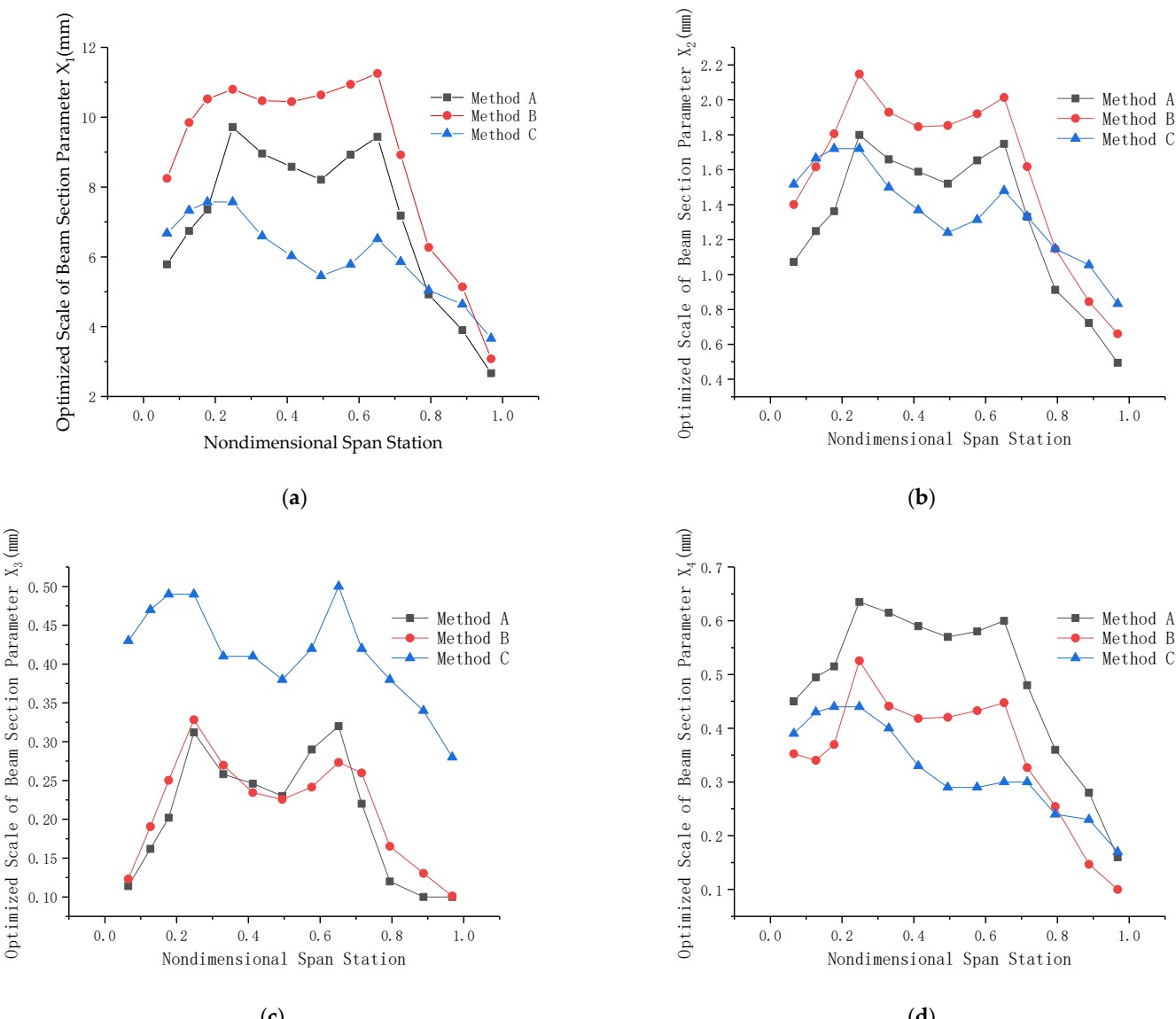

**Figure 14.** Optimized beam section parameters of the front wing and the outer wing. (**a**) Optimized parameter $X_1$; (**b**) optimized parameter $X_2$; (**c**) optimized parameter $X_3$; (**d**) optimized parameter $X_4$.

Figure 16 shows two objective results; one is from the optimization using a traditional exponential function, whereas the other is from the optimization using a modified exponential/linear mixed function. We can see that the modified function results in a much lower value for the objective result, which means that when obtaining the same stiffness level, less weight is needed using the new function for optimization.

**Figure 15.** Optimized beam section parameters of the rear wing. (**a**) Optimized parameter $X_1$; (**b**) optimized parameter $X_2$; (**c**) optimized parameter $X_3$; (**d**) optimized parameter $X_4$.

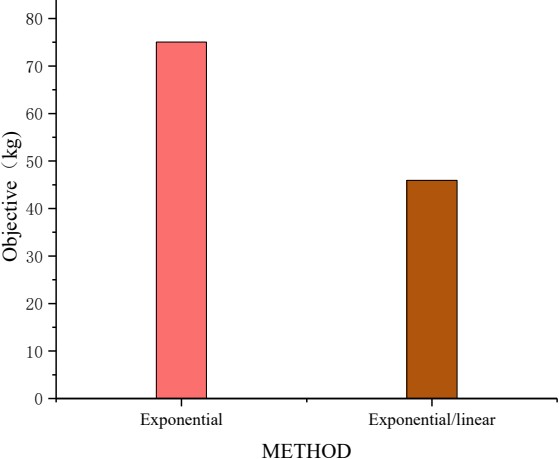

**Figure 16.** Objective optimization using traditional exponential and exponential/linear functions.

### 4.3. Results of the Optimized Stiffness Distribution

To test the accuracy of Methods A and B, the optimized stiffness distribution, including the vertical bending stiffness, the horizontal bending stiffness, and the torsional stiffness of both the front wing and the rear wing were calculated and the results are shown in Figures 17 and 18.

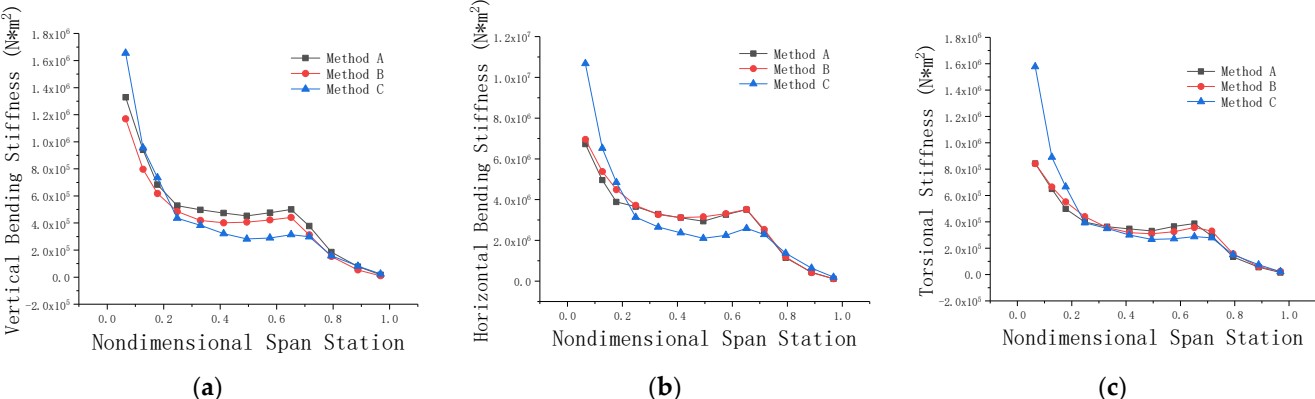

**Figure 17.** Various kinds of stiffness distribution of the front wing optimized by different methods. (**a**) Vertical bending stiffness; (**b**) horizontal bending stiffness; (**c**) torsional stiffness.

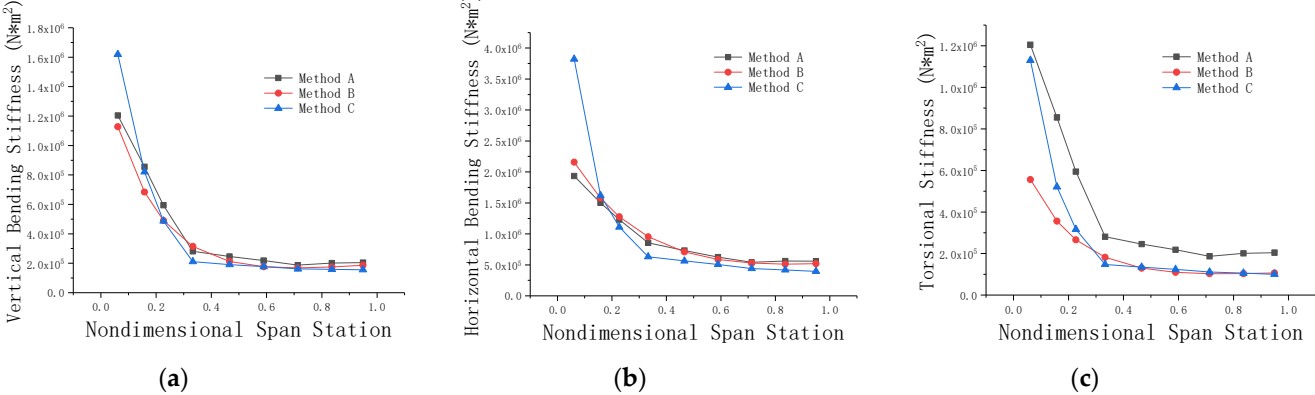

**Figure 18.** Various kinds of stiffness distribution of the rear wing optimized by different methods. (**a**) Vertical bending stiffness; (**b**) horizontal bending stiffness; (**c**) torsional stiffness.

We can see from the figures that the stiffness distributions obtained using the two methods have good consistency and are similar to the distribution of the three-dimensional model except for the first span station point. The reason for this error is that the first point is near the root of the wing, where the ratio of the wingspan to the width of the wing box is relatively small; therefore, the theoretical calculation of an engineering beam is not accurate. Furthermore, the rigid section assumption is adopted for the three-dimensional model, and the wings of the three-dimensional model are connected to the fuselage through rigid multipoint constraint elements. This results in a stiffness increase at the wing root. However, the results have little influence on the overall stiffness distribution and can be ignored in the preliminary design; thus, the methods are valuable for engineering applications. In practical applications, it should be noted that the design parameters of method A and method C are cross-section parameters of the wing, which are relatively intuitive and easy to adjust, whereas the design parameters of method B are parameters of mathematical expressions of stiffness, which do not directly relate to the structure, so this requires conversion between the stiffness and the actual structure.

## 5. Conclusions

In this study, two beam-frame model methods and one three-dimensional model method are proposed for the global stiffness design and optimization of joined wings. A beam-frame model of a joined wing, as an example for use during the preliminary design stage, is established. A three-dimensional detailed model is built as a reference and introduced for comparison. Two methods for the beam-frame model, an optimization of the diagonal beam section, and an optimization method of the exponential/linear combination function fit, defined as method A and method B, respectively, are used. The method for the three-dimensional model is a traditional optimization method based on a genetic algorithm, and is defined as method C. The static aeroelastic responses are calculated and compared under the same flight conditions at 0.2 Ma and 25 km above sea level. The main conclusions are as follows:

1.  The stiffness distributions of the beam-frame model obtained from method A and method B have good consistency and are similar to the distribution of the three-dimensional model of method C;
2.  The stiffness distribution data and the beam section parameters obtained using method B can be converted into parameter sizes of the beam section and conform with the trend of the other methods;
3.  Method B (the optimization method of the exponential/linear combination function fit) obtains a greater weight reduction than the traditional method (the optimization method of the exponential function fit) under the same stiffness conditions;
4.  Compared with method C and other detailed-model-based methods, methods A and B show high computational efficiency and are easy to implement with variable parametrical analysis of the joined wing.

In our future work, we aim to perform joint connection analysis, including the influence of fixed connections, hinged connections, and others, especially the gap of the joint between the front wing and rear wing. The gap nonlinearity of the joined wing will express many interesting problems and will have great research value.

**Author Contributions:** Conceptualization, X.L., X.W., Z.W. and C.Y.; methodology, X.L. and Z.W.; software, X.L.; validation, X.L.; formal analysis, X.L.; investigation, X.L.; resources, X.L.; data curation, X.L.; writing—original draft preparation, X.L.; writing—review and editing, X.L., X.W. and Z.W.; visualization, X.L.; supervision, X.W. and C.Y.; project administration, X.L.; funding acquisition, Z.W. All authors have read and agreed to the published version of the manuscript.

**Funding:** Supported by Zhejiang Key laboratory of General Aviation Operation technology (General Aviation Institute of Zhejiang JianDe), NO: JDGA2020-4.

**Institutional Review Board Statement:** Not applicable.

**Informed Consent Statement:** Not applicable.

**Data Availability Statement:** Not applicable.

**Conflicts of Interest:** The authors declare no conflict of interest.

## Appendix A

As shown in Figure 2a.

$$
\begin{aligned}
I_{z1} &= 2\left(\frac{1}{12}X_3\left(X_6 - \frac{X_1+X_2}{2}\right)^3 + A(X_1)\left(\frac{X_6}{2}\right)^2 + A(X_2)\left(\frac{X_6}{2}\right)^2 + A(X_4)\left(\frac{X_6}{2}\right)^2\right) \\
&= \frac{1}{2}A_t X_6^2 + \frac{1}{6}A(X_3)\left(L_w^2 - 3X_6^2\right) = \frac{1}{2}A_t X_6^2 + \frac{X_3(-\overline{X}^3 - 2X_6^3 + 3X_6\overline{X}^2)}{6}
\end{aligned} \tag{A1}
$$

$$
\begin{aligned}
I_{y1} &= 2\left(\frac{1}{12}X_4\left(X_5 - \frac{X_1+X_2}{2}\right)^3 + A(X_1)\left(\frac{X_5}{2}\right)^2 + A(X_2)\left(\frac{X_5}{2}\right)^2 + A(X_3)\left(\frac{X_5}{2}\right)^2\right) \\
&= \frac{1}{2}A_t X_5^2 + \frac{1}{6}A(X_4)\left(L_s^2 - 3X_5^2\right) = \frac{1}{2}A_t X_5^2 + \frac{X_4(-\overline{X}^3 - 2X_5^3 + 3X_5\overline{X}^2)}{6}
\end{aligned} \tag{A2}
$$

$$I_{t1} = \frac{4\Omega^2}{\oint \frac{ds}{\delta}} = \frac{4(X_5 X_6)^2}{2(\frac{1}{X_1}X_1 + \frac{1}{X_3}L_w + \frac{1}{X_2}X_2 + \frac{1}{X_4}L_s)} = \frac{4(X_5 X_6)^2}{4 + 2(\frac{X_6}{X_3} + \frac{X_5}{X_4}) - (X_1 + X_2)(\frac{1}{X_3} + \frac{1}{X_4})}$$
$$\approx \frac{2(X_5 X_6)^2}{\frac{X_6}{X_3} + \frac{X_5}{X_4}} = \frac{2X_3 X_4 (X_5 X_6)^2}{X_4 X_6 + X_3 X_5} \tag{A3}$$

The first item is directly taken for calculation in the program calculation about $I_{t1}$

$$I_{yz1} = 2\left(A(X_1)\cdot\left(-\frac{X_5 X_6}{4}\right) + A(X_2)\cdot\left(\frac{X_5 X_6}{4}\right)\right) = \frac{X_5 X_6}{2}(A(X_2) - A(X_1)) \tag{A4}$$

where:

$$A(X_1) = X_1^2, \quad A(X_2) = X_2^2, \quad A(X_3) = X_3 L_w, \quad A(X_4) = X_4 L_s$$
$$A_t = \sum_{i=1}^{4} A(X_1), \quad L_s = X_5 - \overline{X}, \quad L_w = X_6 - \overline{X}, \quad \overline{X} = \frac{X_1 + X_2}{2} \tag{A5}$$

In general, the flange size is larger than the size of the web thickness and skin thickness, and as mentioned earlier, to simulate the material distribution, the main diagonal flange size is larger than the secondary diagonal flange size in Figure 2a, so the relations of the cross section parameters are as in Equation (A6):

$$\max(X_3, X_4) < X_2 < X_1 \tag{A6}$$

According to the rotation axis formula of the moment of inertia:

$$I_u = \frac{I_z + I_y}{2} + \frac{I_z - I_y}{2}\cos 2\alpha - I_{yz}\sin 2\alpha \tag{A7}$$

After Taylor expansion of the sine and cosine expression and discarding of the high-order small quantities, the moment of inertia of the beam section relative to the bending axis is:

$$I_{u1} \approx \frac{I_{z1} + I_{y1}}{2} + (1 - 2\alpha^2)\frac{I_{z1} - I_{y1}}{2} - 2\alpha I_{yz1}$$
$$= (1 - \alpha^2)I_{z1} + \alpha^2 I_{y1} - 2\alpha I_{yz1} \tag{A8}$$

For the case of Figure 2b:

$$I_{z2} = I_{z1} \quad I_{y2} = I_{y2} \quad I_{yz2} = 0 \tag{A9}$$

then

$$I_{u2} - I_{u1} = 2\alpha I_{yz1} < 0 \tag{A10}$$

For the case of Figure 2c, the four flange areas are equal ($X = X_1 = X_2$), keeping parameters $X_3$–$X_6$ and $A_t$ unchanged, according to Equations (A1)–(A8):

$$I_{y3} = \frac{1}{2}A_t X_5^2 + \frac{X_4(-X^3 - 2X_5^3 + 3X_5 X^2)}{6} \tag{A11}$$

$$I_{z3} = \frac{1}{2}A_t X_6^2 + \frac{X_3(-X^3 - 2X_6^3 + 3X_6 X^2)}{6} \tag{A12}$$

$$I_{yz3} = 0 \tag{A13}$$

$$I_{u3} = (1 - \alpha^2)I_{z3} + \alpha^2 I_{y3} - 2\alpha I_{yz3} \tag{A14}$$

$$\Delta I_u = I_{u1} - I_{u3} = (1 - \alpha^2)(I_{z1} - I_{z3}) + \alpha^2(I_{y1} - I_{y3}) - 2\alpha I_{yz1} \tag{A15}$$

According to Equations (A1) and (A12):

$$I_{z1} - I_{z3} = \frac{X_3(-\overline{X}^3 + 3X_6\overline{X}^2)}{6} - \frac{X_3(-X^3 + 3X_6 X^2)}{6}$$
$$= \frac{X_3 X_6(\overline{X}^2 - X^2)}{2} + \frac{X_3(-\overline{X}^3 + X^3)}{6} \tag{A16}$$

According to Equations (A2) and (A11):

$$
\begin{aligned}
I_{y1} - I_{y3} &= \frac{X_4(-\overline{X}^3 + 3X_5\overline{X}^2)}{6} - \frac{X_4(-X^3 + 3X_5X^2)}{6} \\
&= \frac{X_4X_5(\overline{X}^2 - X^2)}{2} + \frac{X_4(-\overline{X}^3 + X^3)}{6}
\end{aligned}
\tag{A17}
$$

Substituting Equations (A4), (A16) and (A17) into Equation (A15), we obtain:

$$
\begin{aligned}
\Delta I_u &= (1 - \alpha^2)\left(\frac{X_3X_6(\overline{X}^2 - X^2)}{2} + \frac{X_3(-\overline{X}^3 + X^3)}{6}\right) + \\
&\quad \alpha^2\left(\frac{X_4X_5(\overline{X}^2 - X^2)}{2} + \frac{X_4(-\overline{X}^3 + X^3)}{6}\right) - 2\alpha\frac{X_5X_6}{2}(A(X_2) - A(X_1)) \\
&= \frac{X_3X_6(\overline{X}^2 - X^2)}{2} + \alpha X_5X_6(A(X_1) - A(X_2)) - \frac{X_3(\overline{X}^3 - X^3)}{6} + \\
&\quad \alpha^2\left(\frac{(X_4X_5 - X_3X_6)(\overline{X}^2 - X^2)}{2} + \frac{(X_3 - X_4)(\overline{X}^3 - X^3)}{6}\right)
\end{aligned}
\tag{A18}
$$

Because the scales of the fixed parameters, e.g., $X_5$ and $X_6$, are far greater than the design parameters, e.g., $X_1$–$X_4$, etc., and the tilted angle $\alpha$ is small, the last two terms of Equation (A18) are relatively small compared with the first two terms and can be rounded off; thus:

$$
\begin{aligned}
\Delta I_u &= I_{u1} - I_{u3} = \alpha X_5X_6(A(X_1) - A(X_2)) + \frac{X_3X_6}{2}(\overline{X}^2 - X^2) \\
&= X_6(\alpha X_5(X_1^2 - X_2^2)) + X_3(\overline{X}^2 - X^2)
\end{aligned}
\tag{A19}
$$

$$
\begin{aligned}
&\text{let } X_1 = kX_2, \ k > 1 \\
&\text{then } \overline{X} = \frac{(k+1)}{2}X_2 \\
&\Delta I_u = X_6\left[\alpha X_5(k^2 - 1)X_2^2 + X_3\left(\frac{(k+1)^2}{4}X_2^2 - X^2\right)\right]
\end{aligned}
\tag{A20}
$$

Because $X_3$–$X_6$ and $A_t$ are equal, according to Equation (A5):

$$
\begin{aligned}
&X_1^2 + X_2^2 - \frac{X_1 + X_2}{2}(X_3 + X_4) = 2X^2 - X(X_3 + X_4) \\
&\Rightarrow (k^2 + 1)X_2^2 - \frac{k+1}{2}X_2(X_3 + X_4) = 2X^2 - X(X_3 + X_4) \\
&\Rightarrow X_2 = \frac{\frac{k+1}{2}(X_3 + X_4) + \sqrt{\left(\frac{k+1}{2}(X_3 + X_4)\right)^2 + 4(k^2 + 1)(2X^2 - X(X_3 + X_4))}}{2(k^2 + 1)}
\end{aligned}
\tag{A21}
$$

$$
\begin{aligned}
&\text{let } X_3 + X_4 = 2k_2X, \quad 0 < k_2 < 1 \\
&\Rightarrow X_2 = \frac{(k+1)k_2 + \sqrt{((k+1)k_2)^2 + 8(k^2 + 1)(1 - k_2))}}{2(k^2 + 1)}X
\end{aligned}
\tag{A22}
$$

$0 < k_2 < 1$, because the flange width is greater than the thickness of skin and the web. Let

$$
k_N = \frac{(k+1)k_2 + \sqrt{((k+1)k_2)^2 + 8(k^2 + 1)(1 - k_2))}}{2(k^2 + 1)}
\tag{A23}
$$

then

$$
X_2 = k_N X
\tag{A24}
$$

and

$$
k_N > \frac{(k+1)k_2 + \sqrt{((k+1)k_2)^2}}{2(k^2 + 1)} = \frac{(k+1)k_2}{k^2 + 1} > \frac{(k+1)k_2}{(k+1)^2} = \frac{k_2}{(k+1)}
\tag{A25}
$$

To compare the magnitude of the moment of inertia between them, according to Equation (A20), it can be seen that

$$
\begin{aligned}
\Delta I_u &= X_6[\alpha X_5(k^2-1)k_N^2 + X_3(\tfrac{(k+1)^2}{4}k_N^2 - 1)]X^2 \\
&> X_6 X^2\left\{\alpha X_5(k^2-1)(\tfrac{k}{k+1})^2 + X_3[\tfrac{(k+1)^2}{4}(\tfrac{k}{k+1})^2 - 1]\right\} \\
&= X_6 X^2(\alpha X_5(k-1)\tfrac{k^2}{k+1} + X_3(\tfrac{k^2}{4}-1))
\end{aligned}
\tag{A26}
$$

In order to make $\Delta I_u > 0$, and considering $X_6 X^2 > 0$, then

$$
\alpha X_5(k-1)\frac{k_2^2}{k+1} + X_3(\frac{k_2^2}{4}-1) > 0
\tag{A27}
$$

Let

$$
X_5 = NX_3
\tag{A28}
$$

Bringing Equation (A28) into Equation (A27), then

$$
(\alpha\frac{(k-1)k_2^2}{k+1}N + \frac{k_2^2}{4}-1)X_3 > 0
\tag{A29}
$$

According to Equation (A25), we obtain

$$
N > \frac{(1-\frac{k_2^2}{4})(k+1)}{(k-1)k_2^2\alpha} \quad (k>1, 0<k_2\le 1)
\tag{A30}
$$

**Appendix B**

In Appendix B, the approximate formula for inertias in Figure 2a will be derived. Substituting Equation (A5) into (A1) and expanding it gives

$$
\begin{aligned}
I_{z1} &= \tfrac{1}{2}A_t X_6^2 + \frac{X_3(-\overline{X}^3 - 2X_6^3 + 3X_6\overline{X}^2)}{6} \\
&= \tfrac{1}{2}(X_1^2 + X_2^2 + X_3(X_6 - \tfrac{X_1+X_2}{2}) + X_4(X_5 - \tfrac{X_1+X_2}{2}))X_6^2 + \\
&\quad \frac{X_3(-\overline{X}^3 - 2X_6^3 + 3X_6\overline{X}^2)}{6} \\
&= \tfrac{1}{6}(X_3 X_6 + 3X_4 X_5 + 3((X_1^2 + X_2^2) - \tfrac{X_3+X_4}{2}(X_1 + X_2)))X_6^2 + \\
&\quad \tfrac{1}{6}X_3(-\overline{X}^3 + 3X_6\overline{X}^2)
\end{aligned}
\tag{A31}
$$

According to Equations (A6) and (A20), let

$$
\begin{aligned}
X_1 &= kX_2, \quad k>1 \\
X_2 &= k_3\tfrac{X_3+X_4}{2}, \quad k_3>1
\end{aligned}
\tag{A32}
$$

The beams are thin-walled structures, which means

$$
\overline{X} = \frac{X_1+X_2}{2} << (X_5, X_6)
\tag{A33}
$$

Considering Equation (A33), the last term of Equation (A32) can be omitted; then, substituting Equation (A32) into (A31) gives

$$
I_{z1} \approx \frac{1}{6}(X_3 X_6 + 3X_4 X_5 + 3k_A X_2^2)X_6^2
\tag{A34}
$$

where

$$
k_A = \frac{k_3(k^2+1) - k - 1}{k_3}
\tag{A35}
$$

Considering $k > 1$, $k_3 > 1$, then

$$
\begin{aligned}
k_A &= \frac{k_3(k^2+1)-k-1}{k_3} > \frac{k_3(k+1)-k-1}{k_3} \\
&= \frac{(k_3-1)(k+1)}{k_3} > 0
\end{aligned}
\tag{A36}
$$

Substituting Equation (A5) into (A2), by means of a derivation similar to (A31)–(A34), we obtain

$$
I_{y1} \approx \frac{1}{6}(X_4 X_5 + 3 X_3 X_6 + 3 k_A X_2^2) X_5^2
\tag{A37}
$$

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
