# Peer review of "Aeroelastic Optimization Design of the Global Stiffness for a Joined Wing Aircraft"

_applsci, doi:10.3390/app112411800_

Round 1

Reviewer 1 Report

The article is written in a clear and understandable style.  An introduction describes the essence of the problem and previous methods of solving, and it also includes a sufficient number of references to related works.  The results of the considered paper are clearly presented with series of figures and tables. The conclusions are written succinctly and concretely.

A connected wing includes twin wings arranged in the form of a rhombus both in the top view and in the front view, but the fuselage of airplane  is the connecting diagonal of these wings. This design has a number of advantages,  in particular, it is higher rigidity, and others. However, the wings form builds  a complex system, so stiffness, weight distribution and aerodynamic characteristics must be analyzed compared to a traditional wing. The article provides an analysis of the main characteristics of this system and indicates the range of their possible deviations.

The wing is divided into 3 sections, the front wing, the outer wing and the rear wing. Then the main beam of each partition is divided into several sections. The structural model of the docked wing at the preliminary design stage consists of 270 nodes and 367 elements, etc.  Three methods are used to solve the problem of aeroelastic optimization of an articulated wing, including two methods for determining the stiffness of the beam-frame and one for a three-dimensional model. Equation (20) describes method A. In the method B, the stiffness distributions are optimized in accordance with the exponential function, and improved formula (22) is proposed instead of the traditional one (21). Method C is applied to a three-dimensional model according to reference [2]. The analysis of system elements based on calculations of moments of inertia is carried out.  There two equations are used, the static aeroelastic equation for displacement vector and  flatter analysis equation.

Results indicate that the selected methods are suitable for optimizing and provide high results for stiffness of the wing construction. The Tables 1 and 2 indicate that the constraints are executed well and the results of all the methods used coincide well with each other.  The article can be published in the journal.

There is one my remark that should  be corrected, the word ‘expression’  should be replaced to the word ‘equation’ (224 row and etc).

Reviewer 2 Report

Very good quality work.  The recommendation is that the conclusions should be referred to the practical and industrial application of research and mathematical modeling that would justify the granting of financial research funds.

Reviewer 3 Report

Dear authors,

Please see the attached file for my review of your manuscript. 

Please note that  all comments are of qualitative nature. 
